# Iron in Nepheline: Crystal Chemical Features and Petrological Applications

Julia A. Mikhailova [1,*], Sergey M. Aksenov [1,2], Yakov A. Pakhomovsky [1], Bertrand N. Moine [3], Camille Dusséaux [4], Yulia A. Vaitieva [2] and Mikhail Voronin [5]

1 Geological Institute, Kola Science Centre, Russian Academy of Sciences, 14 Fersman Str., 184209 Apatity, Russia

2 Laboratory of Arctic Mineralogy and Material Sciences, Kola Science Centre, Russian Academy of Sciences, 14 Fersman Str., 184209 Apatity, Russia

3 Laboratoire Magmas et Volcans, Université Clermont Auvergne, CNRS, IRD, OPGC, F-63000 Clermont-Ferrand, France

4 ISTerre, Université Grenoble Alpes, CNRS, 38000 Grenoble, France

5 Institute of Experimental Mineralogy, Russian Academy of Sciences, 4 Academica Osypyana Str., 142432 Chernogolovka, Russia

* Correspondence: mikhailova@geoksc.apatity.ru; Tel.: +7-81555-79333

**Abstract:** Nepheline is a nominally anhydrous aluminosilicate that typically contains an impurity of ferric iron replacing aluminum in tetrahedral sites. However, previous researchers noted the constant presence of ferrous iron in the chemical composition of nepheline from the rocks of the Khibiny and Lovozero massifs (Kola Peninsula, Russia). We have carried out microprobe, spectroscopic, chemical and crystal chemical studies of nepheline from the Lovozero massif. We have established the presence of molecular water in nepheline, and also that the incorporation of ferrous iron into nepheline crystal structure is associated with the simultaneous increasing of the coordination number from four to five (or six) due to the inclusion of the 'additional' water molecules that form point $[FeO_4(H_2O)_n]$-defects (where $n = 1, 2$) in the tetrahedral framework. The nepheline iron content is closely related to the presence of small needle-like aegirine inclusions. The total iron content in nepheline saturated with aegirine needles is approximately an order of magnitude lower than in nepheline free from aegirine inclusions. Most likely the aegirine inclusions in nepheline are formed as a result of the decomposition of the nepheline–"iron nepheline" solid solution. We propose that this process is triggered by the oxidation of ferrous iron in the crystal structure of nepheline.

**Keywords:** nepheline; aegirine; ferrous iron; ferric iron; crystal structure; oxidation

## 1. Introduction

Nepheline with idealized formula $Na_3K[Al_4Si_4O_{16}]$ is a key mineral of many silica-undersaturated igneous rocks and related pegmatites (e.g., Lovozero and Khibiny massifs at Kola Peninsula, Russia; Ilímaussaq in Greenland [1–7]), but can be also found in metamorphosed magmatic ejecta (e.g., Somma-Vesuvius volcanic complex, Italy [8,9]). After bauxite, nepheline rocks are the second most important type of aluminum raw materials [10]. Nepheline is widely used in ceramics, leather, rubber, textiles, wood, and the oil industry [11–13].

The crystal structure of nepheline was solved by Buerger [14,15] and Hahn and Buerger [16]. Later, the structure was refined many times based on both natural and synthetic samples [17–22]. The crystal structure of nepheline is a derivative of tridymite-type framework, where the voids are filled with Na and K atoms, and half of the Si atoms are replaced by Al. The entry of a smaller Na cation into the tridymite framework causes a reduction in size and change in the shape of the rings around the sodium cations. This is achieved by shifting the oxygen atom O(1) by 0.3Å from the threefold axis and rotating

the $T(1)$ and $T(2)$ tetrahedra, which leads to the doubling of the tridymite-type unit cell and the appearance of four crystallographically independent $T(1)$–$T(4)$ sites. Potassium atoms are located in the large pseudohexagonal channels ($A$ sites), whereas the Na atoms occupy small oval-shaped sites ($B$), resulting in an ideal composition of $Na_3K[Al_4Si_4O_{16}]$. In natural samples, the $A$ site is typically two-thirds occupied by K, while one-third of the $A$ site remains vacant ($\square$) due to the entry into the structure an excess amount of Si. The $B$ site is usually almost fully occupied by Na.

Ideally, Al and Si are ordered over the tetrahedral sites in nepheline. The $T(1)$ and $T(4)$ sites are characterized by the predominance of Al, and the $T(2)$ and $T(3)$–by Si. In natural samples of nepheline, a different degree of Si–Al ordering in tetrahedra is observed. The reasons for Si–Al ordering/disordering are questionable [17], but many researchers believe that the degree of ordering depends on the of cooling history of nepheline-containing rocks [23–25].

The strong interest in nepheline arises from the presence of satellite reflections in the diffraction patterns from nepheline crystals in addition to the Bragg reflections. The presence of satellite reflections indicates incommensurate modulation in nepheline crystal structure. The temperature-dependent evolution of satellite reflections has been studied in detail in [20,26]. A study on the evolution of the intensities of the satellite reflections under hydrostatic pressure has shown that these reflections disappear between 1 and 1.8 GPa [27]. Different explanations have been proposed to be at the origin of the modulation in nepheline: ordering of the cation vacancies in the large, pseudohexagonal channel [26,28]; formation of domains with different Al/Si ordering patterns [29]; displacive modulation of the essentially rigid tetrahedra of the framework [30]; coupling between the displacive modulation of the tetrahedral framework and the ordering of cations and vacancies in the pseudohexagonal channel [20].

Nepheline is an important petrological indicator of rock-forming processes. In addition to the fact that the Si–Al ordering of nepheline is related to the thermal history of rocks, the chemical composition of nepheline is also widely used to estimate the crystallization temperature of nepheline-bearing associations. Hamilton proposed a geothermometer which is based on the ratio of alkalis (in form of $Na[AlSiO_4]$ and $K[AlSiO_4]$ end-members) and excess of $SiO_2$ in a solid solution of nepheline [31,32]. However, the calculation of $SiO_2$ excess, and, consequently, the temperature estimates, is significantly affected by the presence of various impurities in the composition of nepheline. According to the works of Henderson [33,34], the calculated excess of silica values is significantly affected when the coupled substitution $2Al^{3+} \leftrightarrow M^{2+} + Si^{4+}$ (where $M^{2+}$ is a small divalent cation) is taking into account. In addition, different impurities in the composition of nepheline reflect the composition of the mineral-forming medium and physicochemical conditions of the crystallization, which expands the possibilities of using nepheline as a petrological indicator. In particular, Mg-rich nepheline occurs in the groundmass of strongly $SiO_2$-undersaturated, feldspar-free, mafic volcanic rocks (olivine-rich foidites) [35]. Hence, the occurrence of Mg-rich nepheline seems to be related to its derivation from Mg-rich magmas.

Impurities in the composition of nepheline are very diverse (among them Ca, Mg, Fe, Ti, Ga, Li, Rb, Cs, Ba, Sr, rare earth elements), and moreover, nepheline also contains $H_2O$ and $CO_2$ [33,36]. Natural nephelines are quite common enriched by iron [33,37,38]. The presence of ferric iron (in the form as $Fe^{3+}$ substituting for $Al^{3+}$ in the tetrahedral sites) is well studied for the framework silicates and it is usually reported as such in nepheline microprobe analyses. However, "wet" chemistry data indicate that, in addition to ferric iron, ferrous iron is constantly present in the composition of nepheline from different rocks of the Lovozero massif [39]. Ferrous iron-bearing nepheline was also evidenced in the rocks of the Khibiny massif [6]. In some nepheline samples, ferrous iron may predominate.

According to previous research [40], the iron content in nepheline from the Lovozero massif is independent from the type of rocks in which nepheline is found. Instead, surprisingly, the content of iron in nepheline is directly related to the presence of micron-scale needle-like inclusions of aegirine inside nepheline grains [40,41]. The total iron content in

aegirine needles-saturated nepheline is approximately an order of magnitude lower than in nepheline free from aegirine inclusions.

In this paper, we discuss two issues related to the presence of iron in nepheline from the Lovozero massif. Firstly, we propose a model for ferrous iron incorporation into the crystal structure of nepheline, and secondly, we discuss the possible reasons for the formation of needle-like aegirine crystals inside nepheline and some petrogenetic features and applications.

## 2. Short Geological and Petrography Backgrounds

The Lovozero alkaline massif (Figure 1a) is a Devonian layered laccolith located in the southwest of the Kola Peninsula (Russia) among Archean gneisses [2,42,43]. It is the world's second largest alkaline massif, with an area of 650 km$^2$. The main rocks of the Lovozero massif are nepheline syenites (leuco- and melanocratic) and foidolites, mainly ijolite and urtite. The bulk of the massif consists of many regularly repeating subhorizontal layers (or rhythms) with the following sequence of rocks (from top to bottom): melanocratic nepheline syenite, leucocratic nepheline syenite, foidolite [2,42]. Transitions between different rocks within a same rhythm are gradual, while contacts between rhythms are sharp. All rhythms together make up the so-called "Layered complex" over a thickness of 1700 m. The Eudialyte complex overlies the Layered complex and consists of melanocratic nepheline syenite enriched in eudialyte-group minerals, co-called eudialyte lujavrite. Nepheline, along with alkali feldspar, alkaline pyroxenes, and amphiboles, is the main rock-forming mineral of most of the rock varieties of the massif. The modal content of nepheline can reach 90 vol.%, for example, in urtite of the Layered complex (Figure 2).

There are two varieties of nepheline in the rocks of the Lovozero massif, which are easily distinguished in thin sections: nepheline without inclusions (Figure 2a; point LV-00-16 in Figure 1b; further described as "pure") and nepheline with numerous inclusions of thin needle-like aegirine crystals (Figure 2b; point LV-335E in Figure 1b). Due to the presence of numerous aegirine inclusions, nepheline has a greenish color and weak magnetic properties [39]. While aegirine is usually evenly distributed within nepheline, some grains show a zonal distribution of inclusions. For example, inclusions of aegirine can be located only in the thin marginal zone of a nepheline grain.

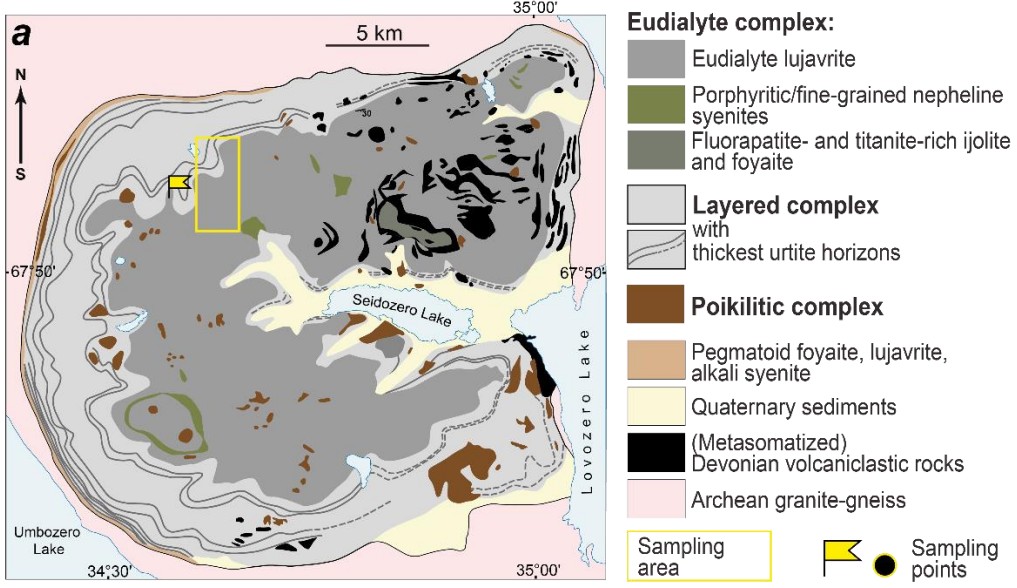

**Figure 1.** *Cont.*

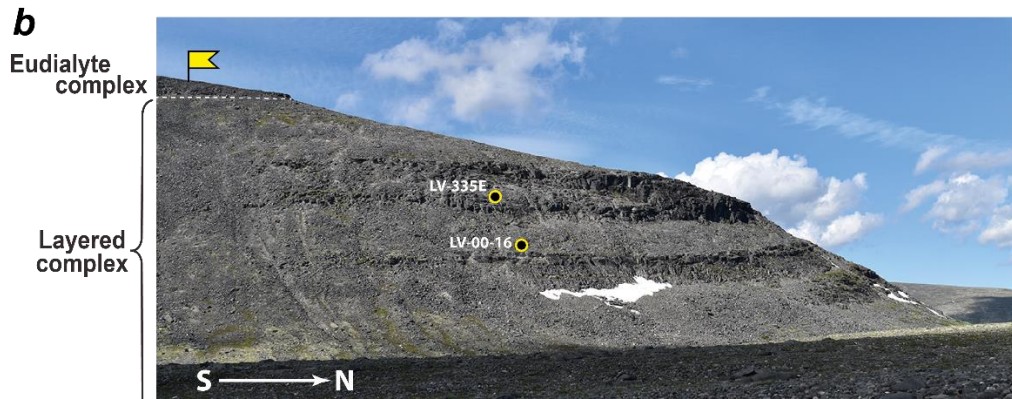

**Figure 1.** Geological background. (**a**) Geological scheme of the Lovozero alkaline massif after [42]; (**b**) urtite sampling points in two neighboring rhythms of the Layered complex. Individual rhythms are clearly visible due to the fact that melanocratic nepheline syenites (the top of each rhythm) are more resistant to weathering.

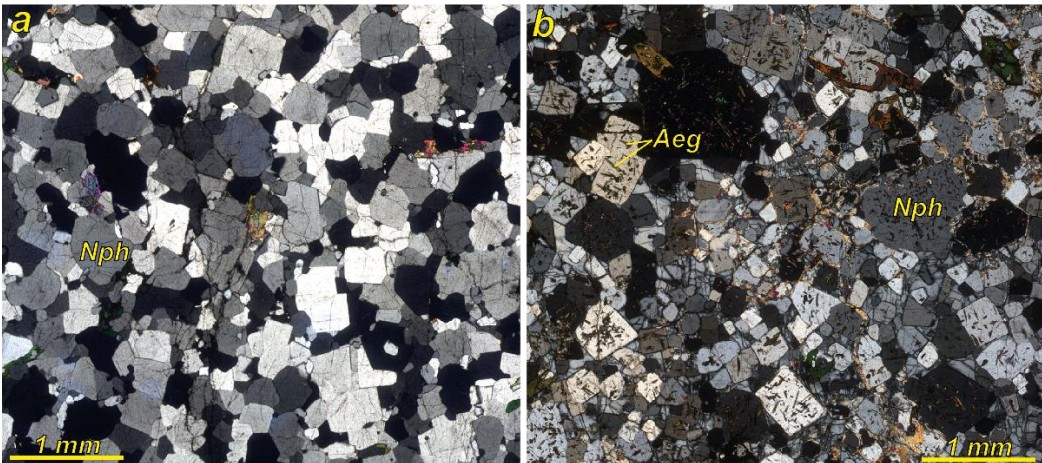

**Figure 2.** Petrography background: two types of nepheline (Nph) found in urtite. (**a**) "pure" nepheline grains containing no inclusions of aegirine; (**b**) nepheline with numerous inclusions of thin needle-like aegirine (Aeg) crystals. Photos of the polished sections in transmitted light.

According to previous research [40], pure nepheline and nepheline containing aegirine inclusions differ significantly in iron content. Namely, the total iron content in nepheline saturated with aegirine needles is approximately an order of magnitude less than in pure nepheline. In the case when aegirine inclusions are present only in the marginal zone of nepheline, there is zoning in the composition of nepheline. There is an order of magnitude more iron in the center of such a nepheline grain than in the marginal zone [44]. Based on works [40,44], we selected materials for this study.

### 3. Materials and Methods

For detailed crystal chemical studies, we chose two samples of urtite (LV-00-16 and LV-335E) from neighboring rhythms of the Layered complex (Figure 1b) that both contain 75–80 vol.% of nepheline (Figure 3a,b). Both urtite samples also contains aegirine (located outside of nepheline grains and usually forming poikilitic crystals), microcline-perthite, magnesioarfvedsonite, loparite-(Ce), a eudialyte group mineral, sodalite, and natrolite. In sample LV-00-16, nepheline does not contain aegirine needle-like inclusions (Figure 3a,c), while nepheline from sample LV-335E is saturated with numerous aegirine needle-like inclusions (Figure 3b,d). The composition of nepheline was determined by microprobe and wet chemistry methods, and the crystal structure of nepheline was solved in both samples. Microprobes analyses also allowed the determination of aegirine composition.

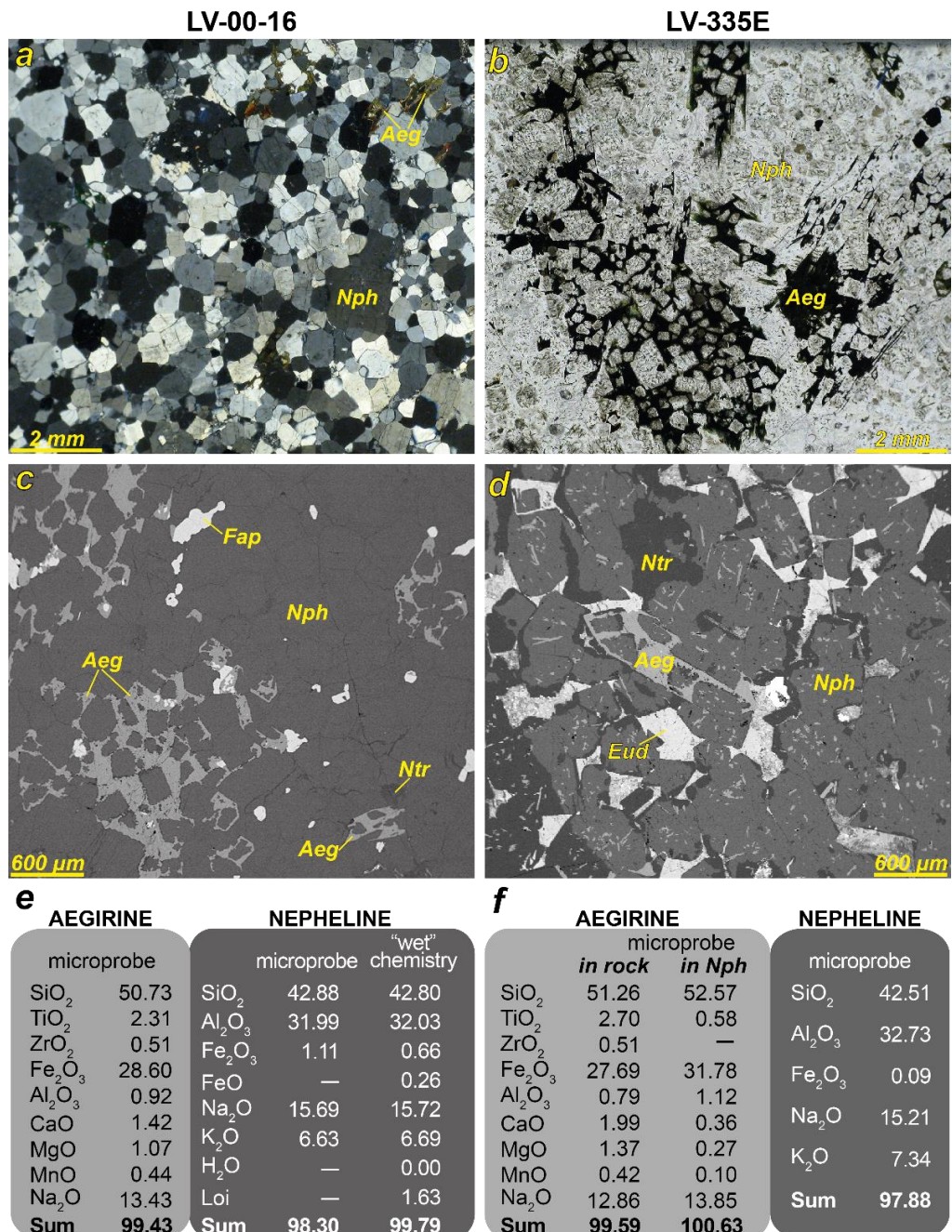

**Figure 3.** Microphotograph of the studied samples and chemical composition of nepheline and aegirine. (**a**) Photo of sample LV-00-16 in polarized light; nepheline does not contain aegirine needle-like inclusions. Aegirine is present only in the surrounding groundmass; (**b**) photo of sample LV-335E in transmitted light; nepheline grains are saturated with fine needle-like crystals of aegirine. Aegirine is also present in the groundmass as large poikilitic crystals; (**c**) and (**d**) are BSE-images of samples LV-00-16 and LV-335E, respectively; (**e**) and (**f**) are the chemical composition of aegirine and nepheline in samples LV-00-16 and LV-335E, respectively. Loi—loss on ignition.

To enable a statistical comparison of the chemical composition of aegirine inside nepheline crystals and aegirine outside nepheline, 35 samples of nepheline syenites and foidolites from the Kedykvyrpakh loparite underground mine (sampling area in Figure 1a) were selected. In these samples, the compositions of nepheline and aegirine were determined only by the microprobe method.

Microprobe analyses of nepheline and aegirine were performed at the Geological Institute, Kola Science Center of the Russian Academy of Sciences (GI KSC RAS, Apatity, Russia) using the Cameca MS-46 electron microprobe (Cameca, Gennevilliers, France) operating in the WDS-mode at 22 kV with a beam diameter of 10 μm, beam current of 20–40 nA, and counting times of 10 s (for a peak) and 10 s (for background before and after the peak), with 5–10 counts for every element in each point. The following standards were used: lorenzenite (Na, Ti), pyrope (Al), wollastonite (Si, Ca), wadeite (K), $MnCO_3$ (Mn), hematite (Fe), $ZrSiO_4$ (Zr). The analytical precision (reproducibility) of mineral analyses was 0.2–0.05 wt% (2 standard deviations) for the major element and approximately 0.01 wt% for impurities. The systematic errors were within the random errors. Back-scattered electron (BSE) images were obtained using a scanning electron microscope LEO-1450 (Carl Zeiss Microscopy, Oberkochen, Germany) with the energy-dispersive system Aztec Ultimmax 100 (Oxford Instruments, UK).

Wet chemical analysis of nepheline was carried out at the GI KSC RAS. The accuracy limits for $SiO_2$, $Fe_2O_3$, FeO, $Al_2O_3$, CaO, $Na_2O$, $K_2O$, $H_2O$ are 0.01 wt%. For chemical analysis, a weighed portion of 20 mg of nepheline was taken. At the beginning of the chemical analysis, nepheline was dissolved in weak HCl. The insoluble residue was removed and the composition of the solution was analyzed. This made it possible to completely exclude the presence of aegirine in the analyzed sample.

Single-crystal X-ray diffraction data were collected at room temperature using an Rigaku XtaLAB Synergy-S diffraction diffractometer with monochromatized radiation Mo$K\alpha$ ($\lambda$ = 0.71073 Å) and a Hybrid Pixel Array detector in the ω-scanning mode. A semi-empirical absorption correction based on the intensities of equivalent reflections was applied, and the data were adjusted for Lorentz effects, polarization, and background using CrysAlis software [45]. Refinement of the unit cell parameters was also performed using the CrysAlis software [45]. The definition and refinement of the structure were carried out using the Jana 2006 software package [46]. The illustrations were created using the JANA2006 software package in combination with the DIAMOND program [47]. The atomic scattering coefficients for neutral atoms, together with corrections for anomalous dispersion, were taken from International Tables of Crystallography [48].

Fourier-transform infrared spectroscopy (FTIR) spectra were obtained at the Laboratoire Magmas et Volcans (LMV; Université Clermont Auvergne, France). Nepheline FTIR spectra were acquired on double-polished 200 μm-thick sections on a Bruker Vertex 70 FTIR (Fourier-transform infra-red spectroscope) coupled with a Hyperion microscope equipped with ×15 objective and condenser at LMV. Beam size in the analyses varied from 30 to 50 μm. The spectra were collected through a $CaF_2$ plate with a resolution of 2 cm$^{-1}$ and with up to 300 scans.

Mössbauer spectroscopy turned out to be inapplicable for the study of iron in nepheline due to the presence of rare inclusions of aegirine, even in pure nepheline.

Mineral abbreviations [49], and corresponding mineral names and formulas are shown in Table 1.

**Table 1.** Mineral abbreviations.

| Abbreviation | Mineral | Formula * |
|---|---|---|
| Aeg | aegirine | $NaFe^{3+}Si_2O_6$ |
| Eud | eudialyte-group mineral | [50] |
| Fap | fluorapatite | $Ca_5(PO_4)_3F$ |
| Lop-Ce | loparite-(Ce) | $(Na,Ce,Sr)(Ce,Th)(Ti,Nb)_2O_6$ |
| Marf | magnesio-arfvedsonite | $NaNa_2(Fe^{2+}_4Fe^{3+})Si_8O_{22}(OH)_2$ |
| Nph | nepheline | $Na_3K(Al_4Si_4O_{16})$ |
| Ntr | natrolite | $Na_2(Si_3Al_2)O_{10}·2H_2O$ |

*—mineral formulas are given in accordance with IMA (International Mineralogical Association) list of minerals, with the exception of eudialyte-group mineral.

## 4. Results

*4.1. Chemical Composition of Nepheline and Aegirine (Samples LV-00-16 and LV-335E)*

The chemical composition of pure nepheline (sample LV-00-16) and nepheline saturated with inclusions of aegirine, as well as aegirine located inside nepheline and aegirine in the rock (sample LV-335E) was obtained by microprobe analysis. Additionally, the composition of pure nepheline (sample LV-00-16) was also determined by wet chemistry. The results are presented in Figure 3.

Nepheline without aegirine inclusions contains an order of magnitude more iron than nepheline saturated with aegirine inclusions (Figure 3e,f), with a third of this iron being ferrous. According to microprobe analysis, the formula for pure nepheline (based on 16 oxygen atoms) is as follows:

$Na_{2.99}K_{0.83}Al_{3.70}Fe^{3+}_{0.08}Si_{4.21}O_{16}$ (sample LV-00-16),

while the formula for nepheline saturated with inclusions is:

$Na_{2.81}K_{0.92}Al_{3.80}Fe^{3+}_{0.01}Si_{4.19}O_{16}$ (sample LV-335E).

The formula for pure nepheline, calculated based on the results of wet chemical analysis, is as follows:

$Na_{3.00}K_{0.84}Al_{3.72}Fe^{3+}_{0.05}Fe^{2+}_{0.03}Si_{4.21}O_{16}$ (sample LV-00-16)

and matches the formula calculated from the data of microprobe analysis.

The sum of microprobe analysis of pure nepheline is very low for a nominally anhydrous mineral. Based on a comparison of microprobe and wet chemistry data, the lack of the sum is mainly relates to molecular water, which is removed when the nepheline is heated.

The composition of aegirine in the groundmass differs significantly from the composition of aegirine inside nepheline grains. Nepheline contained inclusions of pure (Na-Fe) aegirine include a small admixture of a jadeite (Na-Al) end-member, whereas aegirine in the groundmass is relatively enriched in the diopside component as well as titanium and zirconium (Figure 3e,f). Accordingly, the formulas of aegirine (based on 4 cations and 6 oxygen atoms) are as follows:

1.　aegirine in rock (LV-00-16): $Na_{1.00}\,Ca_{0.06}\,Mg_{0.06}\,Mn_{0.02}\,Fe^{3+}_{0.83}\,Al_{0.04}\,Ti_{0.07}\,Si_{1.95}\,O_{6}$;
2.　aegirine in rock (LV-335E): $Na_{0.96}\,Ca_{0.08}\,Mg_{0.08}\,Mn_{0.01}\,Fe^{3+}_{0.80}\,Al_{0.04}\,Ti_{0.08}\,Si_{1.96}\,O_{6}$;
3.　aegirine inside nepheline (LV-335E): $Na_{1.02}\,Ca_{0.01}\,Mg_{0.01}\,Fe^{3+}_{0.91}\,Al_{0.05}\,Ti_{0.02}\,Si_{1.99}\,O_{6}$.

*4.2. Chemical Composition of the Nepheline and Aegirine (Samples from Kedykvyrpakh Loparite Deposit)*

Representative microprobe analyses of nepheline and aegirine from nepheline syenite and foidolites of the Kedykvyrpakh deposit are presented in Table 2, and all data on the chemical composition of these minerals can be found in Supplementary Table S1.

The iron content in nepheline from the studied samples varies from 0 to 0.03 atoms per formula unit (*apfu*; median 0.01 *apfu*). The chemical composition of aegirine located inside the nepheline grains differ significantly from aegirine located outside nepheline grains. Figure 4 shows the results of data factor analysis on the composition of aegirine from the studied samples. Aegirine, located inside nepheline, is relatively enriched in aluminum (median 0.09 *apfu*). Aegirine, located outside the nepheline crystals, is enriched in calcium (median 0.11 *apfu*) and magnesium (median 0.09 *apfu*), as well as titanium (median 0.07 *apfu*), zirconium (median 0.01 *apfu*), and manganese (median 0.02 *apfu*).

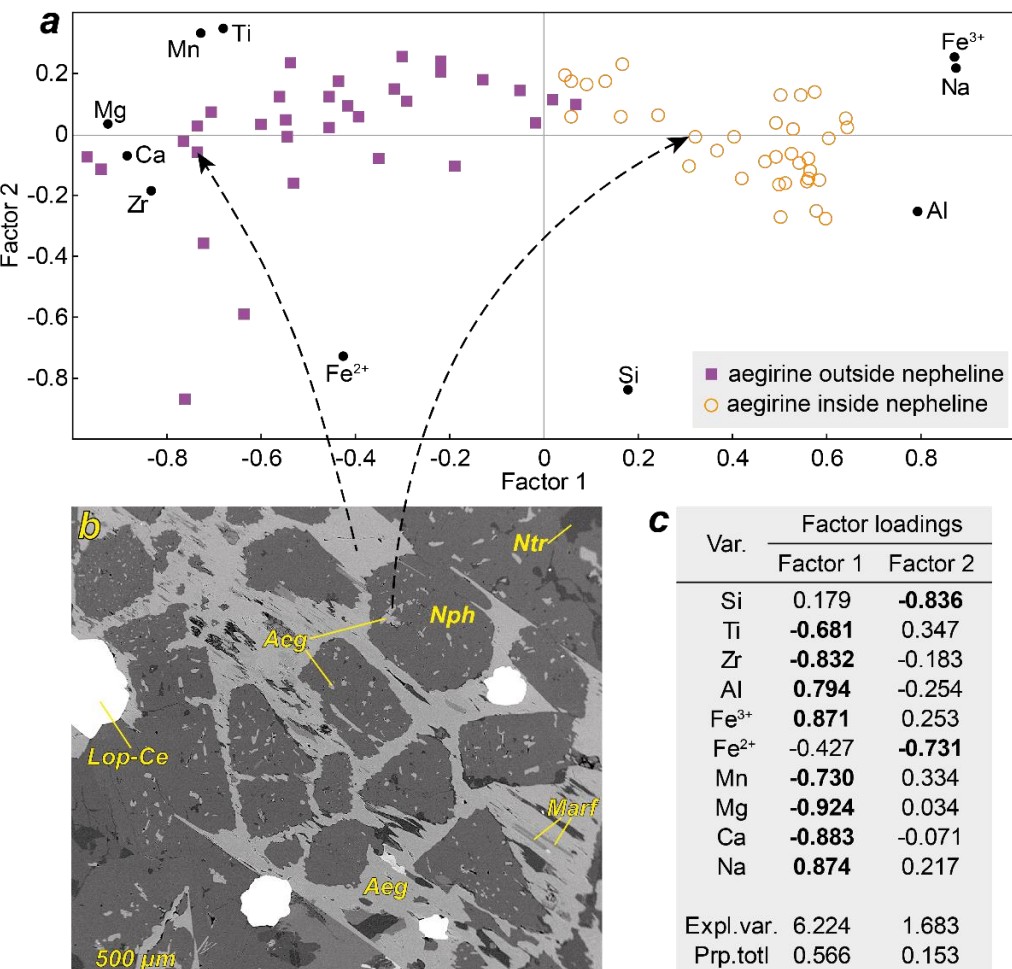

**Figure 4.** A visual representation of variations in the chemical composition of aegirine. (**a**) Results of data factor analyses of aegirine composition; aegirine located inside the nepheline grains differ in chemical composition from aegirine located outside nepheline grains; (**b**) BSE-image of urtite sample LV-III-5-6; the composition of aegirine inside and outside of nepheline is different (see also Table 2). Dashed arrows connect the point of analysis and the point corresponding to the chemical composition; (**c**) table of factor loadings. Var.—variables; Expl.var.—Explained variance; Prp.totl—proportion of total variance. Factor loadings > |0.5| are shown in bold.

**Table 2.** Representative chemical analyses of nepheline and aegirine in Lovozero urtite (see text for explanation).

| Sample | | LV-III-5-6 | | | LV-III-5-2 | | | LV-III-4-5 | |
|---|---|---|---|---|---|---|---|---|---|
| Mineral | Nph | Aeg outside Nph | Aeg inside Nph | Nph | Aeg outside Nph | Aeg inside Nph | Nph | Aeg inside Nph | Aeg outside Nph |
| $SiO_2$ | 42.45 | 52.83 | 52.64 | 40.95 | 52.17 | 53.05 | 42.89 | 53.60 | 53.72 |
| $ZrO_2$ | – | 0.59 | b.d.l. | – | 0.57 | b.d.l. | – | b.d.l. | b.d.l. |
| $TiO_2$ | – | 4.16 | 2.96 | – | 2.31 | 0.44 | – | 2.67 | 3.44 |
| $Al_2O_3$ | 33.53 | 0.96 | 1.24 | 33.51 | 0.99 | 2.32 | 32.95 | 1.18 | 1.29 |
| FeO | 0.13 | 24.42 | 24.99 | 0.12 | 23.01 | 27.04 | 0.13 | 26.08 | 24.65 |
| MgO | – | 1.22 | 0.93 | – | 2.88 | 0.45 | – | 0.83 | 0.82 |
| CaO | – | 2.09 | 0.90 | – | 5.15 | 0.13 | – | 0.54 | 0.48 |
| MnO | – | 0.56 | 0.42 | – | 0.60 | 0.05 | – | 0.47 | 0.88 |
| $Na_2O$ | 16.21 | 13.18 | 14.81 | 15.89 | 11.17 | 14.11 | 16.68 | 14.45 | 14.58 |
| $K_2O$ | 7.21 | – | – | 7.22 | – | – | 6.61 | – | – |
| Sum | 99.52 | 100.06 | 99.02 | 97.69 | 98.84 | 97.59 | 99.25 | 99.83 | 99.94 |

**Table 2.** *Cont.*

| Sample | LV-III-5-6 | | | LV-III-5-2 | | | LV-III-4-5 | | |
|---|---|---|---|---|---|---|---|---|---|
| Mineral | Nph | Aeg outside Nph | Aeg inside Nph | Nph | Aeg outside Nph | Aeg inside Nph | Nph | Aeg inside Nph | Aeg outside Nph |
| Formula based on 16 oxygens for nepheline and on 4 cations and 6 oxygens for aegirine, *apfu* | | | | | | | | | |
| Si | 4.13 | 1.96 | 1.94 | 4.06 | 1.96 | 1.98 | 4.17 | 1.97 | 1.97 |
| Ti | – | 0.12 | 0.08 | – | 0.07 | 0.01 | – | 0.07 | 0.09 |
| Zr | – | 0.01 | 0.00 | – | 0.01 | 0.00 | – | 0.00 | 0.00 |
| Al | 3.84 | 0.04 | 0.05 | 3.92 | 0.04 | 0.10 | 3.78 | 0.05 | 0.06 |
| $Fe^{3+}$ | 0.01 | 0.74 | 0.77 | 0.01 | 0.70 | 0.85 | 0.01 | 0.80 | 0.75 |
| $Fe^{2+}$ | – | 0.02 | 0.00 | – | 0.02 | 0.00 | – | 0.00 | 0.00 |
| Mn | – | 0.02 | 0.01 | – | 0.02 | 0.00 | – | 0.01 | 0.03 |
| Mg | – | 0.07 | 0.05 | – | 0.16 | 0.03 | – | 0.05 | 0.04 |
| Ca | – | 0.08 | 0.04 | – | 0.21 | 0.01 | – | 0.02 | 0.02 |
| Na | 3.05 | 0.95 | 1.06 | 3.06 | 0.81 | 1.02 | 3.15 | 1.03 | 1.03 |
| K | 0.89 | – | – | 0.91 | – | – | 0.82 | – | – |
| Sum | 11.92 | 4.00 | 4.00 | 11.96 | 4.00 | 4.00 | 11.92 | 4.00 | 4.00 |

b.d.l.—below detection limit; *apfu*—atoms per formula unit.

### 4.3. FTIR Spectroscopy

The spectrum of nepheline shows absorption bands in the region of the fundamental vibration of OH-group and molecular water (3000–3800 cm$^{-1}$). The spectrum displays vibrations $\nu_1 = 3489$ cm$^{-1}$ and $\nu_3 = 3551$ cm$^{-1}$. All these bands are shifted toward lower wave numbers compared with the parameters of an isolated and noninteracting water molecule, $\nu_1 \approx 3650$ cm$^{-1}$ and $\nu_3 \approx 3760$ cm$^{-1}$ (Figure 5a,b).

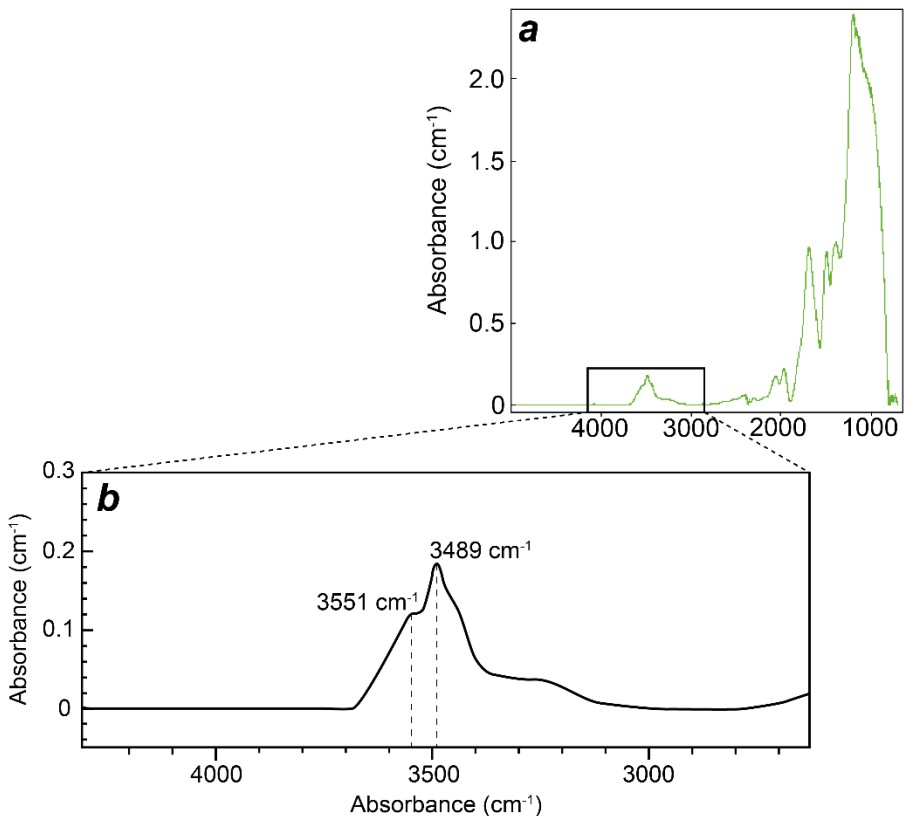

**Figure 5.** Unpolarized FTIR spectra of nepheline. (**a**) In region 1000–5000 cm$^{-1}$; (**b**) in the region of the fundamental vibration of OH-group and molecular water (3000–3800 cm$^{-1}$). Sample LV-01-4, Kedykvyrpakh loparite deposit.

### 4.4. Crystal Structure of the Nepheline

The following hexagonal unit-cell parameters have been obtained: $a = 9.9965(7)$ Å, $c = 8.3796(17)$Å, $V = 725.19(16)$ Å$^3$ for sample LV-00-16 and $a = 10.0020(3)$ Å, $c = 8.3831(2)$ Å, $V = 726.29(4)$ Å$^3$ for sample LV-335E. In accordance with the analysis of systematic absence of reflections the space group $P6_3$ was chosen. The experimental details of the data collection and refinement results are listed in Table 3.

**Table 3.** Crystal parameters, data collection and structure refinement details for the crystals of nepheline from the Lovozero alkaline massif.

| Sample Number | LV-00-16 | LV-355E |
|---|---|---|
| Mineral name | Nepheline | Nepheline |
| Formula weight (g) | 570.6 | 575.4 |
| Temperature (K) | 293 | |
| Cell setting | Hexagonal | |
| Space group | $P6_3$ | |
| $a$ (Å) | 9.9965(7) | 10.0020(3) |
| $b$ (Å) | 9.9965(7) | 10.0020(3) |
| $c$ (Å) | 8.3796(17) | 8.3831(2) |
| $V$ (Å$^3$) | 725.19(16) | 726.29(4) |
| $Z$ | 2 | |
| Calculated density, $D_x$ (g cm$^{-3}$) | 2.613 | 2.6504 |
| Crystal size (mm) | $0.02 \times 0.06 \times 0.08$ | $0.01 \times 0.03 \times 0.06$ |
| Crystal form | Irregular grain | Irregular grain |
| Data collection | | |
| Diffractometer | Rigaku XtaLAB Synergy, HyPix detector | |
| Radiation; $\lambda$ | MoK$_\alpha$; 0.71073 | |
| Absorption coefficient, $\mu$ (mm$^{-1}$) | 1.045 | 1.109 |
| $F$ (000) | 561 | 561 |
| Data range $\theta$(°); $h, k, l$ | 3.38–33.55; $-12 < h < 0$, $0 < k < 15$, $-12 < l < 12$ | 3.38–33.15; $-15 < h < 15$, $-11 < k < 15$, $-11 < l < 12$ |
| No. of measured reflections | 1692 | 8368 |
| Total reflections ($N_2$)/observed($N_1$) | 1623/1551 | 1602/1510 |
| Criterion for observed reflections | $I > 2\sigma(I)$ | |
| $R_{int}$ (%) | 0.015 | 0.016 |
| Refinement | | |
| Refinement on | Full-matrix least squares on $F$ | |
| Weight scheme | $1/(\sigma^2 |F| + 0.0004F^2)$ | $1/(\sigma^2 |F| + 0.0004F^2)$ |
| $R_1, wR_1 [I > 2\sigma(I)]$ | 0.0191, 0.0282 | 0.0211, 0.0328 |
| $R_1, wR_1$ [all] | 0.0204, 0.0286 | 0.0233, 0.0334 |
| GooF (Goodness of fit) | 1.12 | 1.27 |
| Max./min. residual $e$ density, ($e$Å$^{-3}$) | 0.27/−0.27 | 0.24/−0.30 |

The distribution of the cations over crystallographic site was performed in accordance with the previously described methodology [17], taking into account the scattering factors at the sites, interatomic distances and ionic radii of cations. The final refinement cycles converged with: $R_1 = 0.0191$, w$R_1$ (all) = 0.0266, GOF = 1.12 for 1551 $I > 2\sigma(I)$ for sample LV-00-16; $R_1 = 0.0211$, w$R_2 = 0.0334$, GOF = 1.27 for 1510 $I > 2\sigma(I)$ for sample LV-335E. The deepest minimum and highest peak in the final residual electron density map were: 0.27 $e$Å$^{-3}$ and $-0.27$ $e$Å$^{-3}$ for sample LV-00-16; 0.24 $e$Å$^{-3}$ and $-0.30$ $e$Å$^{-3}$ for sample LV-335E. CCDC 2,194,355 and 2,194,356 contain the supplementary crystallographic data for these compounds. The data can be obtained free of charge from The Cambridge Crystallographic Data Centre via www.ccdc.cam.ac.uk/structures. Bond valence sums (BVS) calculations were performed using the bond length parameters from [51,52].

The crystal structure of nepheline is characterized by a tetrahedral framework of a tridymite-type topology with the presence of wide channels of two types filled by extra-

framework *A* and *B* type cations. The general formula of nepheline can be written as ($Z$ = 2): |$AB_3$|[$T(1)T(2)T(3)_3T(4)_3O_{16}$], where *A* cations occupy the channels with the regular hexagonal ring in cross-section, while *B* cations occupy the channels with the oval hexagonal rings in cross-section. The symmetry of the tridymite-type framework (the aristotype) is characterized by the space group *P6$_3$/mmc*; however, different types of the cation ordering both in tetrahedral and extra-framework sites lead to the lowering of the symmetry to the space group *P6$_3$* (typical for the most samples of nepheline). Moreover, the complex schemes of the cation ordering also lead to formation of modulated structures with the lower symmetry and multiplied unit cell parameters [53].

The crystal structures of the studied samples are generally similar to the previously studied ones with the space group *P6$_3$* [17,18,24,54–57] (Figure 6). The refined crystal chemical formulas are ($Z$ = 2): |$^A$(K$_{0.72}$□$_{0.28}$)$^B$(Na$_{2.898}$□$_{0.102}$)|[Fe$_{0.08}$Al$_{3.538}$Si$_{4.382}$O$_{16}$] for the sample LV-00-16 and |$^A$(K$_{0.842}$□$_{0.158}$)$^B$(Na$_{2.868}$□$_{0.132}$)|[Al$_{3.797}$Si$_{4.203}$O$_{16}$] for the sample LV-335E. The distribution of the elements over crystallographic sites and their occupancies are listed in Table 4.

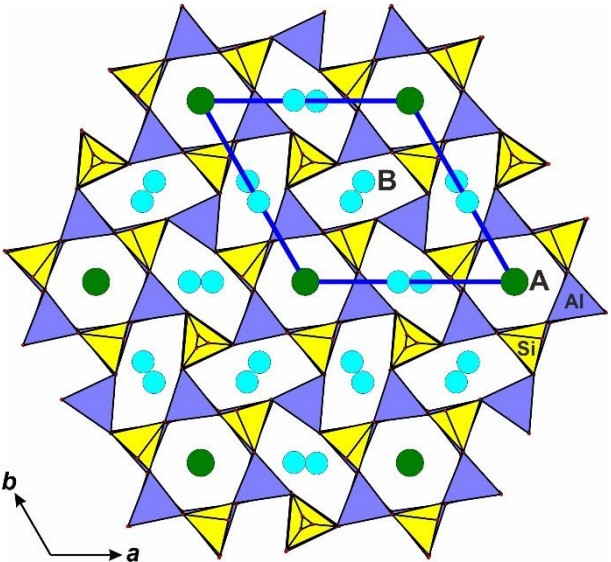

**Figure 6.** General view of the crystal structure of nepheline.

**Table 4.** Site composition, mean distances and amounts of electrons ($e_{calc}$), electrons per formula unit) in the crystal structures of studied samples of nepheline from the Lovozero alkaline massif.

| Site | LV-00-16 | | | LV-355E | | |
|------|----------|----------|-------------|---------|----------|-------------|
|      | $e_{calc}$ | Mean Distance | Composition | $e_{calc}$ | Mean Distance | Composition |
| A    | 13.49 | 3.011 | K$_{0.72}$□$_{0.28}$ | 16.00 | 3.007 | K$_{0.842}$□$_{0.158}$ |
| B    | 11.54 | 2.620 | Na$_{2.898}$□$_{0.102}$ | 10.52 | 2.624 | Na$_{2.868}$□$_{0.132}$ |
| T(1) | 13.0 | 1.721 | Al$_{0.72}$Si$_{0.26}$Fe$_{0.02}$ | 12.42 | 1.701 | Al$_{0.66}$Si$_{0.34}$ |
| T(2) | 13.48 | 1.618 | Si$_{0.89}$Al$_{0.11}$ | 14.00 | 1.637 | Si$_{0.78}$Al$_{0.22}$ |
| T(3) | 13.64 | 1.617 | Si$_{2.662}$Al$_{0.338}$ | 13.45 | 1.621 | Si$_{2.53}$Al$_{0.37}$ |
| T(4) | 12.75 | 1.727 | Al$_{2.37}$Si$_{0.57}$Fe$_{0.06}$ | 12.81 | 1.730 | Al$_{2.46}$Si$_{0.54}$ |

The crystal structure of nepheline contains four independent crystallographic *T*(1-4) sites which form the tetrahedral framework. The studied nepheline samples are both enriched by silicon. Based on the mean distances in $TO_4$ tetrahedra in framework aluminosilicates, the following equation was proposed to determine the amount of Si and Al in each site [58]:

$$y = 6.3481x - 10.178 \tag{1}$$

where $y$ is the amount of Al in the tetrahedral site, and $x$ is the mean distance in $TO_4$ tetrahedron. The O1 site in both samples is disordered around the three-fold axes. However, in the crystal structure of nepheline the additional site located directly at three-fold axes with the coordinates (1/3 2/3 $z$) (where $z \sim 0$) is also observed [24]. Despite of the predominance of Si over Al in the studied samples the analysis of the Fourier maps demonstrated the absence of any considerable peaks of electron density at three-fold axes between split O1 sites.

The channels with the regular hexagonal ring in cross-section are occupied by *A* sites filled by potassium (0.72 *apfu* and 0.842 *apfu* in the samples LV-00-16 and LV-335E, respectively). The *B* site which occupies the channel with the oval cross-section is also characterized by the minor amount of vacancy and the amounts of sodium are 2.898 *apfu* and 2.868 *apfu* in the samples LV-00-16 and LV-335E, respectively. The refined crystal chemical formulas for both samples in the part of Si/Al ratio and partial occupancies of the A and B sites are generally in good agreement with the chemical compositions of the samples. The lower refined amount of potassium from single crystal X ray analysis in comparison with the value obtained by EMPA can be explained by the zonal type of the studied crystals typical for nepheline.

## 5. Discussion

Consistently with previous works [39], our microprobe and wet chemistry dataset shows that iron is present in the chemical composition of nepheline from the rocks of the Lovozero massif. In addition, we evidence the presence of molecular water in nepheline from the Lovozero massif based on spectroscopy data.

The incorporation of ferric iron into the tetrahedral sites of natural and synthetic aluminosilicates and zeolites frameworks of have been studied in detail using different spectroscopic methods [59–64]. Within the felspar family [65], the incorporation of $Fe^{3+}$ into the framework sites has been observed for natural ferrisanidine, $K[Fe^{3+}Si_3O_8]$ [66] as well as different synthetic analogs [67]. The maximal content of $Fe^{3+}$ has been observed in synthetic $Cs[Fe^{3+}SiO_4]$ [68] and $Cs[Fe^{3+}TiO_4]$ [69] with the ABW type framework. Ferric iron is a common impurity in the chemical composition of nepheline. The maximum concentration of iron (8.02 wt% $Fe_2O_3$) was found in nepheline in silicate lava from the Oldoinyo Lengai volcano in Tanzania [70]. In nepheline from other localities, the iron content is lower and varies from 0.05 to 0.10 wt% $Fe_2O_3$ [71,72]. In fact, the presence of ferric iron in nepheline from the rocks of the Lovozero massif is not unusual. However, this study highlights that nepheline from the rocks of the Lovozero and Khibiny massifs, in addition to ferric iron, contains a significant amount of ferrous iron, in agreement with previous works. Unlike ferric iron, the distribution and structural behavior of ferrous iron aluminosilicate solids and glasses remains questionable because of the different possible co-ordinational environment (4-, 5-, and 6-fold coordination) [73,74].

It has previously been demonstrated that small amounts of a hydrous component exist in nepheline from volcanic ejecta and nepheline syenites from a variety of localities [8,36,75,76]. The infrared studies of Beran [75] and Beran and Rossman [77] on nepheline from Mount Somma, Italy, indicated that the chemical species $H_2O$ was crystallographically oriented and was most likely incorporated into the nepheline structure in the potassium site.

The direct incorporation of $Fe^{3+}$ into $T(2)$ and/or $T(3)$-sites with tetrahedral coordination correlates with the similar ionic radii of aluminum and ferric iron. However, ferrous iron is characterized by larger ionic radii compared to aluminum. Therefore, the occupancy of tetrahedral sites by $Fe^{2+}$ does not seems possible due to crystal chemical restrictions.

The possible crystal chemical schemes for the incorporation of $Fe^{2+}$ and $Ti^{4+}$ has been recently proposed for natural and synthetic pollucites [78] with the ANA type tetrahedral framework [79] using the concept of blocky isomorphism (or heteropolyhedral substitutions), which is known for numerous mineralogical groups [80–83]. For compounds $CsTiSi_2O_{6.5}$ [84] and iron analog of pollucite [85], it was proposed that the incorporation

of $Fe^{2+}$ associates with the increasing on the co-ordinational number from four to five (or six) by the insertion of the additional water molecules forming point $[Fe^{2+}O_4(H_2O)n]$-defects (where $n = 1, 2$). The same scheme of blocky isomorphism can be also applied to describe the incorporation of ferrous iron into the nepheline crystal structure (Figure 7). The additional water molecules can be inserted into the channel with the regular hexagonal cross-section where they substitute *A* cations. The approximate coordinates for the oxygen of 'additional' water molecules are (0.0025–0.05 0.1). This water molecule increase the coordination number of the *T*(4) site from 4 to 5, and the resulting mean distance of $[T(4)O_4(H_2O)]$ polyhedron is ~2.37 Å. Similar incorporations of minor amount of water molecules into the channel have been also described for other minerals [86–88].

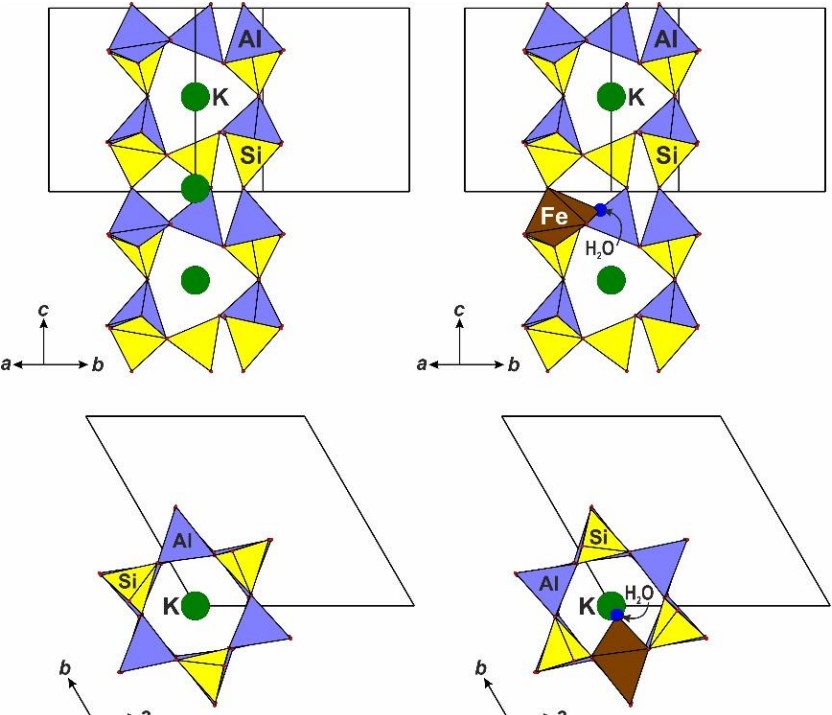

**Figure 7.** Proposed model for the incorporation of ferrous iron into the crystal structure of nepheline with the increasing of coordination number of *T*(4) site by the "additional" water molecules substituted potassium at A site which occupy the channel with the regular hexagonal cross-section.

The content of iron in nepheline is closely related to the presence of aegirine inclusions. Indeed, nepheline from sample LV-00-16 does not contain aegirine inclusions, and the total iron content in this nepheline is 0.92 wt% (Figure 3e). Nepheline from sample LV-335E is saturated with small inclusions of aegirine, and the iron content in this nepheline is 0.09 wt% (Figure 3f). Our results show that the total iron content in nepheline saturated with aegirine needles is approximately an order of magnitude lower than in nepheline free from aegirine inclusions, in agreement with previous studies [40,41]. Nephelines from the Kedykvyrpakh loparite deposit containing aegirine inclusions are depleted in Fe (median 0.11 wt% $Fe_2O_3$, Supplementary Table S1). At the same time, aegirine inside nepheline grains differs in composition from aegirine outside nepheline. According to microprobe analyses, aegirine located inside nepheline is enriched in aluminum, whereas aegirine located outside nepheline contains impurities of calcium, magnesium, titanium, zirconium, and manganese (Table 2, Figure 4).

Another astonishing fact regarding aegirine inclusions-bearing nepheline is that nepheline syenites massifs (e.g., Lovozero and Khibiny) contain elevated concentrations of hydrogen and hydrocarbon gases, mainly methane, localized in (micro)cracks in rocks, as well as in micron-scale secondary inclusions in rock-forming minerals [41,89,90]. In

the Lovozero massif, secondary inclusions are mainly found around aegirine needle-like inclusions in nepheline, whereas "pure" nepheline, without aegirine inclusions, does not contain secondary gas inclusions [41].

In the study of the system nepheline-acmite (aegirine) K. Yagi [91] found that nepheline crystallizes below the solidus temperature, and that it is not associated with other phases between $NaAlSiO_4$ 70–100 wt%. It is suggested that this nepheline might be an iron-bearing solid solution. Bailey and Schairer [92] have shown that nephelines which crystallize from compositions in the system $Na_2O$-$Fe_2O_3$-$Al_2O_3$-$SiO_2$ have higher mean indices of refraction compared with those of iron-free nepheline, indicating the effect of solid solution of the iron-bearing molecule ($NaFe^{3+}SiO_4$ = "iron nepheline"). Onuma and co-authors [37] have proposed a binary phase diagram for the $NaAlSiO_4$-$NaFe^{3+}SiO_4$ join. They reported that nepheline can form a solid solution in the presence of iron where $Al^{3+}$ is substituted by $Fe^{3+}$. Pure "iron nepheline" ($NaFe^{3+}SiO_4$) is an end-member for this join and not exist in isolation as a crystalline phase. Employing X-ray diffraction, Onuma et al. [37] showed that iron-bearing nephelines have larger unit cell dimensions as iron increases in their compositions, which implies that Fe incorporates into the nepheline lattice, as $Fe^{3+}$ has a larger ionic radius than $Al^{3+}$.

If the content of $NaFe^{3+}SiO_4$ in nepheline exceeds 25 molar % (at 700 °C and 1 atm), then "iron nepheline" decomposes according to the scheme [37,92]:

$6NaFe^{3+}SiO_4 = 2NaFeSi_2O_6$ (acmite) + $2Na_2SiO_3$ (sodium metasilicate) + $2Fe_2O_3$ (hematite).

Previously, M. Dorfman and co-authors [44], and S. Ikorsky [41] suggested that inclusions of aegirine inside nepheline are formed as a result of the decomposition of an iron-rich nepheline solid solution. Our data on the chemical composition of aegirine supports this assumption. Iron could be included in nepheline composition in both ferrous and ferric forms during its crystallization. After nepheline crystallization, ferrous iron can be oxidized in accordance with the scheme: $Fe^{2+} + OH^- = Fe^{3+} + O^{2-} + 0.5 H_2$. As a result, the molar content of the $NaFe^{3+}SiO_4$ end-member increases. Thus, the oxidation of ferrous iron can be a trigger for the decomposition of a nepheline–"iron nepheline" solid solution. We propose that aegirine inclusions and closely associated gas inclusions can form in nepheline this way. However, further studies are required to establish the origin of aegirine inclusions in nepheline and provide a better understanding of gas formation and migration in Lovozero.

## 6. Conclusions

1. Nepheline from the rocks of the Lovozero alkaline massif constantly contains ferrous iron in addition to ferric iron. The presence of molecular water in the composition of nepheline was also evidenced.
2. The incorporation of ferrous iron into the nepheline crystal structure is associated with an increase in the coordination number from four to five (or six), due to the inclusion of additional water molecules that form point $[FeO_4(H_2O)n]$-defects (where $n$ = 1, 2).
3. The iron content in nepheline is closely related to the presence of small needle-like aegirine inclusions. The total iron content in nepheline saturated with aegirine needles is approximately an order of magnitude lower than in nepheline free from aegirine inclusions.
4. It is probable that aegirine inclusions in nepheline formed as a result of the decomposition of the nepheline–"iron nepheline" solid solution. This process is triggered by the oxidation of ferrous iron in the crystal structure of nepheline.

**Supplementary Materials:** The following supporting information can be downloaded at: https://www.mdpi.com/article/10.3390/min12101257/s1, Table S1: Chemical composition of nepheline and aegirine.

**Author Contributions:** Conceptualization, J.A.M. and S.M.A.; methodology, B.N.M. and J.A.M.; validation, C.D. and Y.A.P.; investigation, B.N.M., S.M.A., Y.A.V. and M.V.; data curation, Y.A.P.; writing—original draft preparation, J.A.M. and S.M.A.; writing—review and editing, Y.A.P. and C.D.; visualization, J.A.M. and S.M.A. All authors have read and agreed to the published version of the manuscript.

**Funding:** Field work, microprobe, wet chemical, and single-crystal X-ray diffraction measurements were funded by Russian Science Foundation, project no. 21-47-09010; Spectroscopic studies were funded by French National Research Agency (ANR): ANR-20-CE01-0020 H2KOLA; This is Laboratory of Excellence ClerVolc contribution No. 568.

**Data Availability Statement:** Not applicable.

**Acknowledgments:** We are grateful to reviewers who helped us improve the presentation of our results.

**Conflicts of Interest:** The authors declare no conflict of interest.

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
