# Peer review of "Iron in Nepheline: Crystal Chemical Features and Petrological Applications"

_minerals, doi:10.3390/min12101257_

Round 1
Reviewer 1 Report
The authors provided a detailed study of nepheline with different amount of Fe and show the existence of H2O molecules in the generally anhydrous mineral. The manuscript is well written and has to be published in Minerals. Nevertheless, I have some small comments, which can improve the manuscript. Below is a list of suggested corrections:
Lines 40–41: There are a lot of other refinements of the nepheline crystal structure. I suggest changing the sentence: ‘Later, the structure was refined based on both natural and synthetic samples [4–6].’ in a such way: ‘Later, the structure was refined many times based on both natural and synthetic samples.’ And delete the references 4 – 6 or add a couple of more recent ones.
Lines 121–122: Are there any influence on the chemical composition of nepheline, when aegerine inclusions are located in the marginal zone, comparing with nepheline, where aegerine is evenly distributed?
Line 227: It is obviously Figure 3e, f (not 2e,f).
Lines 258–262: Please, provide also the whole FTIR spectrum of nepheline to be sure that there is no additional peaks, related to aegerine.
Line 445: The authors does not mentioned their contributions
Author Response
The authors are very grateful for the high evaluation of the article and important comments. The following changes have been made to the text of the article.
Point 1. Lines 40–41: There are a lot of other refinements of the nepheline crystal structure. I suggest changing the sentence: ‘Later, the structure was refined based on both natural and synthetic samples [4–6].’ in a such way: ‘Later, the structure was refined many times based on both natural and synthetic samples.’ And delete the references 4 – 6 or add a couple of more recent ones.
Response 1. The sentence has been corrected, and references have been replaced. Added the following references:
Tait, K.T.; Sokolova, E.V.; Hawthorne, F.C.; Khomyakov, A.P. The Crystal Chemistry of Nepheline. The Canadian Mineralogist 2003, 41, 61–70, doi:10.2113/gscanmin.41.1.61.
Vulić, P.; Balić-Žunić, T.; Belmonte, L.J.; Kahlenberg, V. Crystal Chemistry of Nephelines from Ijolites and Nepheline-Rich Pegmatites: Influence of Composition and Genesis on the Crystal Structure Investigated by X-Ray Diffraction. Mineral Petrol 2011, 101, 185–194, doi:10.1007/s00710-010-0143-5.
Antao, S.M.; Hassan, I. Nepheline: Structure of Three Samples from the Bancroft Area, Ontario, Obtained Using Synchrotron High-Resolution Powder X-Ray Diffraction. Can Mineral 2010, 48, 69–80, doi:10.3749/canmin.48.1.69.
Angel, R.J.; Gatta, G.D.; Ballaran, T.B.; Carpenter, M.A. The Mechanism of Coupling in the Modulated Structure of Nepheline. Can Mineral 2008, 46, 1465–1476, doi:10.3749/canmin.46.6.1465.
Hovis, G.L.; Crelling, J.; Wattles, D.; Dreibelbis, B.; Dennison, A.; Keohane, M.; Brennan, S. Thermal Expansion of Nepheline-Kalsilite Crystalline Solutions. Mineral Mag 2003, 67, 535–546, doi:10.1180/0026461036730115.
Point 2. Lines 121–122: Are there any influence on the chemical composition of nepheline, when aegerine inclusions are located in the marginal zone, comparing with nepheline, where aegerine is evenly distributed?
Response 2. In the case when aegirine inclusions are present only in the marginal zone of nepheline, there is zoning in the composition of nepheline. There is an order of magnitude more iron in the center of such a nepheline grain than in the marginal zone. This explanation has been added to the text (lines 132-134).
Point 3. Line 227: It is obviously Figure 3e, f (not 2e,f).
Response 3. Corrected.
Point 4. Lines 258–262: Please, provide also the whole FTIR spectrum of nepheline to be sure that there is no additional peaks, related to aegerine.
Response 4. Figure 5 has been modified. Added FTIR spectra of nepheline in the region 1000-5000 cm-1.
Point 5. Line 445: The authors does not mentioned their contributions
Response 5. Authors' contribution has been added.
Reviewer 2 Report
This paper provides a very detailed interesting study on natural nepheline samples. The paper is, overall, very well written. Data are of a very high quality, in particular single XRD data, and the scientific discussion is robust. I don’t see any issues for Minerals to reject the paper, although the authors have to provide more references. I understand that the list is already long, but I believe it is important to recognize the previous contributions published on this topic.
The following improvements have to be consider:
L 36: please provide the references
L37: please provide the references
L39: please add a few references
L40: I think that the authors forget the very first CIF published by Buerger et al. in 1947, please add the reference .
L63: I think the authors should mention the state of the art (at some point, not necessarily here) of the PT studies done on nepheline using DAC and/or multianvil, the system is very studied and the literature is incredibly vast. For instance, the HP study made by Gatta et al. (2007), the multianvil study of Akaogi et al. (2022)
L76 : very diverse(among them, Ca, Mg……..)
L77: The system CaO-MgO-Al2O3-SiO2 is known to be able to host small amount of CO2 and/or H2O, but the authors have to provide at the very least one reference nevertheless.
L103: please recall reference 21 and/or 22
L118: I think it’s better for the reader if you say also here in which sampling site you found pure nepheline (it should be LV0016, Fig 2a). Same thing for the sample with aegirine inclusions (fig 2B, site LV335e).
CIF Aks_LV-00-16: please check the chemical formula in the CIF: I think it’s not the same reported in the paper. Moreover, the Al (t1) s.o.f reported in the CIF is 1.019(7). I think it should be better to fix it to 1.00
CIF Aks_LV-335: please check the chemical formula in the CIF: I think it’s not the same reported in the paper.
I understand that the nepheline that don’t have aegirine needles (LV0016) has about 1 wt% of Fe, but can the author speculate on what is the critical Fe wt% above which we assist to the formation of aegirine (obviously should be more than 1wt% ) ?
Author Response
The authors are very grateful for the high evaluation of the article and important comments. The following changes have been made to the text of the article.
Point 1. L 36: please provide the references
Response 1. The following references have been added:
Markl, G.; Marks, M.; Schwinn, G.; Sommer, H. Phase Equilibrium Constraints on Intensive Crystallization Parameters of the Ilímaussaq Complex, South Greenland. Journal of Petrology 2001, 42, 2231–2258, doi:10.1093/petrology/42.12.2231.
Gerasimovsky, V.I.; Volkov, V.P.; Kogarko, L.N.; Polyakov, A.I.; Saprykina, T.V.; Balashov, Y.A. Geochemistry of the Lovozero Alkaline Massif; Nauka: Moscow, 1966;
Sorensen, H. The Agpaitic Rocks - an Overview. Mineral Mag 1997, 61, 485–498.
Larsen, L.M.; Sørensen, H. The Ilímaussaq Intrusion-Progressive Crystallization and Formation of Layering in an Agpaitic Magma. Geol Soc Spec Publ 1987, 30, 473–488, doi:10.1144/GSL.SP.1987.030.01.23.
Sørensen, H. Brief Introduction to the Geology of the Ilímaussaq Alkaline Complex, South Greenland, and Its Exploration History. Geology of the Greenland Survey Bulletin 2001, 190, 7–23.
Kostyleva-Labuntsova, E.E.; Borutskii, B.Е.; Sokolova, M.N.; Shlykova, Z. v. Mineralogy of the Khibiny Massif: Magmatism and Postmagmatic Transformations; Nauka: Moscow, 1978;
Semenov, E.I. Mineralogy of the Lovozero Alkaline Massif; Nauka: Moscow, 1972;
Point 2. L37: please provide the references
Response 2. The following references have been added:
Balassone, G.; Beran, A. Variable Water Content of Nepheline from Somma-Vesuvio, Italy. Mineral Petrol 1995, 52, 75–83, doi:10.1007/BF01163127.
Balassone, G.; Kahlenberg, V.; Altomare, A.; Mormone, A.; Rizzi, R.; Saviano, M.; Mondillo, N. Nephelines from the Somma-Vesuvius Volcanic Complex (Southern Italy): Crystal-Chemical, Structural and Genetic Investigations. Mineral Petrol 2014, 108, 71–90, doi:10.1007/s00710-013-0290-6.
Point 3. L39: please add a few references
Response 3. The following references have been added:
Sizyakov, V.M.; Bazhin, V.Y.; Sizyakova, E. v. Feasibility Study of the Use of Nepheline-Limestone Charges Instead of Bauxite. Metallurgist 2016, 59, 1135–1141, doi:10.1007/s11015-016-0228-4.
Bagani, M.; Balomenos, E.; Panias, D. Nepheline Syenite as an Alternative Source for Aluminum Production. Minerals 2021, 11, doi:10.3390/min11070734.
Jena, S.K.; Dhawan, N.; Rao, D.S.; Misra, P.K.; Mishra, B.K.; Das, B. Studies on Extraction of Potassium Values from Nepheline Syenite. Int J Miner Process 2014, 133, 13–22, doi:10.1016/j.minpro.2014.09.006.
Point 4. L40: I think that the authors forget the very first CIF published by Buerger et al. in 1947, please add the reference .
Response 4. The following reference has been added:
Buerger, M.J. Derivative Crystal Structures. J Chem Phys 1947, 15, 17–27, doi:10.1063/1.1746278.
Point 5. L63: I think the authors should mention the state of the art (at some point, not necessarily here) of the PT studies done on nepheline using DAC and/or multianvil, the system is very studied and the literature is incredibly vast. For instance, the HP study made by Gatta et al. (2007), the multianvil study of Akaogi et al. (2022)
Response 5. A new paragraph has been added to the text of the article (lines 61-72).
Point 6. L76 : very diverse(among them, Ca, Mg……..)
Response 6: Corrected.
Point 7. L77: The system CaO-MgO-Al2O3-SiO2 is known to be able to host small amount of CO2 and/or H2O, but the authors have to provide at the very least one reference nevertheless.
Response 7: References added.
Point 8. L103: please recall reference 21 and/or 22
Response 8: Corrected.
Point 9. L118: I think it’s better for the reader if you say also here in which sampling site you found pure nepheline (it should be LV0016, Fig 2a). Same thing for the sample with aegirine inclusions (fig 2B, site LV335e).
Response 9: Corrected.
Point 10. CIF Aks_LV-00-16: please check the chemical formula in the CIF: I think it’s not the same reported in the paper.
Response 10. The formula in the CIF differs from the formula in the article because it has been refined in accordance with the methodology proposed in work Jones, J.B. Al–O and Si–O Tetrahedral Distances in Aluminosilicate Framework Structures. Acta Crystallogr B 1968, 24, 355–358, doi:10.1107/S0567740868002360 (lines 324-329).
Point 11. Moreover, the Al (t1) s.o.f reported in the CIF is 1.019(7). I think it should be better to fix it to 1.00
Response 11: Corrected.
Point 12. CIF Aks_LV-335: please check the chemical formula in the CIF: I think it’s not the same reported in the paper.
Response 12. The formula in the CIF differs from the formula in the article because it has been refined in accordance with the methodology proposed in work Jones, J.B. Al–O and Si–O Tetrahedral Distances in Aluminosilicate Framework Structures. Acta Crystallogr B 1968, 24, 355–358, doi:10.1107/S0567740868002360 (lines 324-329).
Point 13. I understand that the nepheline that don’t have aegirine needles (LV0016) has about 1 wt% of Fe, but can the author speculate on what is the critical Fe wt% above which we assist to the formation of aegirine (obviously should be more than 1wt% ) ?
Response 13. One can only make an assumption for a relatively high temperature. According to research K. Onuma and collegues, NaFe3+SiO4 is incorporated into nepheline, by Fe3+ - Al replacement, as much as 25 mol% NaFe3+SiO4 at 700oC at one atmosphere. Thus, the critical concentration of iron in nepheline is 13.37 wt. % Fe2O3 (t = 700oC and p=1 atm).
